# Rapid Increase in the IS*26*-Mediated *cfr* Gene in *E. coli* Isolates with IncP and IncX4 Plasmids and Co-Existing *cfr* and *mcr-1* Genes in a Swine Farm

**DOI:** 10.3390/pathogens10010033

**Published:** 2021-01-03

**Authors:** Zhenbao Ma, Jiao Liu, Lin Chen, Xiaoqin Liu, Wenguang Xiong, Jian-Hua Liu, Zhenling Zeng

**Affiliations:** 1Guangdong Provincial Key Laboratory of Veterinary Pharmaceutics Development and Safety Evaluation, South China Agricultural University, Guangzhou 510642, China; mazhenbao@stu.scau.edu.cn (Z.M.); 20182027012@stu.scau.edu.cn (J.L.); chenl@stu.scau.edu.cn (L.C.); liuxiaoqin@haid.com.cn (X.L.); xiongwg@scau.edu.cn (W.X.); 2National Risk Assessment Laboratory for Antimicrobial Resistance of Animal Original Bacteria, College of Veterinary Medicine, South China Agricultural University, Guangzhou 510642, China; 3Guangdong Laboratory for Lingnan Modern Agriculture, South China Agricultural University, Guangzhou 510642, China

**Keywords:** multi-resistant *cfr* gene, *E. coli*, IS*26*-mediated, co-transfer with *mcr-1*, swine farm

## Abstract

This paper aimed to investigate the molecular epidemiological features of the *cfr* gene in *E. coli* isolates in a typical swine farm during 2014–2017. A total of 617 *E. coli* isolates were screened for the *cfr* gene using PCR amplification. A susceptibility test, pulsed-field gel electrophoresis (PFGE), S1-PFGE, southern blotting hybridization, and the genetic context of the *cfr* gene were all used for analyzing all *cfr*-positive *E. coli* isolates. A conjugation experiment was conducted with the broth mating method using *E. coli* C600 as the recipient strain and 45 *mcr-1-cfr*-bearing *E. coli* isolates as the donor strain. Plasmids pHNEP124 and pHNEP129 were revealed by Illumina Miseq 2500. Eighty-five (13.7%) *E. coli* isolates were positive for the *cfr* gene and the prevalence of the *cfr* gene had significantly increased from 1.6% in 2014 to 29.1% in 2017. The Pulsed-Field Gel Electrophoresis (PFGE) analysis indicated that the spread of the *cfr* gene among *E. coli* isolates was mainly due to horizontal transfer. In addition, the *cfr* gene was primarily located on the plasmids between 28.8-kb to 60-kb in size, and the *cfr* gene was flanked by two copies of IS*26* with the same orientation. Sequence analysis suggested that the plasmids pHNEP124 and pHNEP129 co-harboring the *cfr* and *mcr-1* genes belonged to the plasmids IncP plasmid and IncX4 plasmid, respectively. In conclusion, this is the first study to report the high prevalence of the *cfr* gene among *E. coli* isolates and the first report of the complete genome sequence of IncP and IncX4 plasmids carrying the *mcr-1* and *cfr* genes. The occurrence and dissemination of the *cfr/mcr-1*-carrying plasmids among *E. coli* isolates need further surveillance.

## 1. Introduction

The multi-resistant *cfr* gene, encoded rRNA methyltransferase, confers cross-resistance to five chemically unrelated classes of antimicrobial agents, including phenicols, lincosamides, oxazolidinones, pleuromutilins, and streptogramin A (called PhLOPS_A_ phenotype) [1]. Moreover, it also decreases susceptibility to the 16-membered macrolides spiramycin and josamycin [2]. Since the first identified *cfr* gene in plasmid pSCFS1 from a *Staphylococcus sciuri* isolate, the transferable *cfr* gene has been detected in both Gram-positive and Gram-negative bacteria such as *Staphylococcus*, *Enterococcus*, *Bacillus*, *Macrococcus*, *Jeotgalicoccus*, *Streptococcus*, *Proteus*, *Escherichia coli*, and *Morganella morganii* [3,4,5]. Mobile gene elements such as insertion sequences (ISs) and plasmids can acquire antimicrobial resistance genes and play a vital role in the dissemination of *cfr* in both Gram-negative and Gram-positive genera [4,6]. 

Multidrug-resistant (MDR) *Escherichia coli* has become a worrisome issue that poses a threat to public health and it is also considered to be a major reservoir of antibiotic resistance genes that may be responsible for the treatment failure events in human clinical and veterinary medicine [7]. To date, a total of 24 *cfr*-positive *E. coli* isolates have been reported in food-producing animals from various sources in seven provinces of China and it is mainly located on various plasmid replicon types such as IncX4, IncF43:A-:B-, and F14:A-:B- [8]. Mobile colistin resistance gene *mcr-1* has experienced global dissemination and has spread to over 40 countries or regions covering five continents since the first identification in 2015 [9,10]. Colistin-resistant bacteria has raised serious concern and increased risk to human and animal health because of colistin, which is used as a last-resort drug for treating MDR Gram-negative bacteria infections [11]. Moreover, there are no reports that the *cfr* gene co-exists with the *mcr-1* gene among *E. coli* isolates. A previous study of our lab had reported that the *cfr* gene was been detected in *E. coli* with IncF43:A-:B- plasmid pHNEP28 and *S. sciuri* in a commercial swine farm in 2013 [12]. Thus, in the present study, we continued monitoring the prevalence of the *cfr* gene in *E. coli* isolates from this swine farm during 2014–2017. Interestingly, we found that the prevalence of *cfr* gene had rapidly increased in *E. coli* isolates, and we identified, for the first time, the *cfr* and *mcr-1* genes coexisting in various plasmids such as IncP plasmid and IncX4 plasmid.

## 2. Results

### 2.1. The Prevalence of the cfr Gene and Detection of Other Resistance Genes

As shown in Table 1, a total of 85 (13.7%) *E. coli* isolates were positive for the *cfr* gene, and the prevalence of the *cfr* gene among 617 *E. coli* isolates had significantly increased from 2014 to 2017 (1.6% in 2014, 8.6% in 2015, 19.9% in 2016, and 29.1% in 2017). These strains were isolated from environmental samples including soil (n = 1) and sewage (n = 1), and from anal swab samples of different growth stages of pigs consisting of nursery pigs (n = 29), fattening pigs (n = 52), sows (n = 1), and boars (n = 1), through different stages of growth. Compared with the detectable rate of the *cfr* gene among *E. coli* isolates from different growth stages, the fattening pigs had the highest detectable rate of the *cfr* gene (25.2%), followed by nursery pigs (22.5%), boars (7.7%), and sows (0.6%). No *cfr*-carrying *E. coli* isolates were detected from suckling piglets (Table 1). Moreover, the encoding florfenicol efflux pump gene *floR* was detected in all of the *cfr*-positive *E. coli* isolates, while *fexA*, *fexB*, and *optrA* were not. It is worth noting that 45 (52.9%) *cfr*-positive *E. coli* isolates were positive for the *mcr-1* gene (Figure 2). Compared with the previous reports, which found 24 *cfr*-positive *E. coli* isolates, this study observed that the *mcr-1* gene coexisted in *cfr*-positive *E. coli* isolates with the high detection rate in this study.

### 2.2. Antibiotic Susceptibility Testing 

As shown in Figure 1, susceptibility testing showed that all *cfr*-positive *E. coli* isolates were highly resistant to ampicillin (100%), florfenicol (100%), tetracycline (95.3%), and doxycycline (91.9%), followed by sulfamethoxazole/trimethoprim (87.2%), ciprofloxacin (67.4%), neomycin (64.0%), colistin (52.9%), gentamycin (36.0%), and fosfomycin (33.7%). These isolates showed a lower resistance to apramycin (22.1%), cefotaxime (14.0%), cefquinome (12.8%), cefoxitin (7.0%), amikacin (5.8%), and ceftazidime (2.3%). All *cfr*-positive *E. coli* isolates were susceptible to imipenem. Interestingly, the prevalence of the *mcr-1* gene among *cfr*-positive *E. coli* had decreased sharply from 86.4% before 2017 to 17.1% in 2017, which implies that the results may be a consequence of the ban on colistin as a feed additive for animals in China.

### 2.3. Pulsed-Field Gel Electrophoresis (PFGE) Patterns, Location, and Genetic Context of the cfr Gene

The PFGE analysis revealed that 85 *cfr*-positive *E. coli* isolates were grouped into 49 PFGE patterns, designated A to AW (Figure 2). The PFGE analysis of all *cfr*-bearing *E. coli* isolates showed that most of these isolates were genetically divergent and epidemiologically unrelated. This result suggests that the spread of the *cfr* gene among *E. coli* was mainly due to horizontal transfer. The S1-PFGE and Southern blot hybridization analyses confirmed that the *cfr* gene of 46 *E. coli* isolates was successfully located on plasmids ranging from 28.8 kb to 244.4 kb in size (Figure 2). Two *cfr*-bearing plasmids coexisted in three isolates (GDE7P80, GDE7P92, and GDE7P163). Moreover, the *cfr* gene was mainly distributed on the plasmids in size between 28.8 kb and 60 kb (Figure 2). In addition, the genetic structure of the *cfr* gene was flanked by two copies of IS*26* with the same orientation in 76 *E. coli* isolates and one copy of IS*26* was found to be located upstream of the *cfr* gene in seven *E. coli* isolates (Figure 2). 

### 2.4. Co-Transfer of the mcr-1 and cfr Genes

A conjugative experiment demonstrated that the plasmids carrying the *mcr-1* and *cfr* genes of 30 *E. coli* were successfully transferred to *E. coli* C600 using Luria–Bertani (LB) agar plates containing streptomycin (3000 mg/L) and colistin (2 mg/L). Susceptibility testing indicated that all transconjugants were resistant to colistin, followed by ampicillin (n = 16), sulfamethoxazole/trimethoprim (n = 3), florfenicol (n = 3), apramycin (n = 2), and doxycycline (n = 2). One transconjugant exhibited resistance to fosfomycin and cefotaxime, and one transconjugant exhibited resistance to gentamycin (Table 2). Notably, apart from strains GDE6P133J and GDE6P151J carried in the *floR* gene, the rest of the *cfr*-positive transconjugants showed that the minimum inhibitory concentration (MIC) values of florfenicol and colistin were improved by 0.5–4-fold and 16–32-fold compared with the recipients, respectively. The PCR-based replicon typing (PBRT) analysis indicated that IncX4 plasmid (n = 19) was the most prevalent incompatible (Inc) plasmid type out of the transferable plasmids co-harboring the *mcr-1* and *cfr* genes. Other plasmid types such as IncP, IncI2, and IncHI2 were also been detected in transconjugants. In accordance with the characterization of the *cfr*-positive transconjugants, 12 transformants were obtained and positive for *mcr-1* and *cfr* genes (Appendix A). Susceptibility testing indicated that all *cfr*-positive transformants were susceptible to florfenicol and the MIC values of florfenicol were improved 1–2 folds compared with *E. coli* DH5α. It is noteworthy that all *cfr*-positive transconjugants or transformants except for strains GDE6P133J and GDE6P151J failed to mediated resistance to florfenicol.

### 2.5. Plasmid Analysis of pHNEP124

The sequence analysis revealed that plasmid pHNEP124 is 60430 base pairs (bp) in size with an average GC content of 47.01% and belonged to plasmid pMCR_1511-like IncP type plasmid, which consisted of a typical plasmid backbone and two mosaic variant regions. The backbone of plasmid pHNEP124 is comprised of the *trfA* encoding plasmid replication initiation protein, two par modules for plasmid partitioning, a toxin-antitoxin *higA-B* system, and a “*KlcA-kleE*” region encoding a host-lethal protein for plasmid maintenance and stability, as well as the two conjugative regions *tra* (~17.3-kb) and *trb* (~12.7-kb) for plasmid horizontal transfer. BLASTn analysis indicated that the backbone of the plasmid pHNEP124 shared a high identity (>99%) with those *mcr-1*-carrying IncP type plasmids found in *Enterobacteriaceae*, such as *E. coli* plasmid pHNGDF36-1 (Genbank accession no. MF978389), *Klebsiella pneumoniae* pMCR_1511 (Genbank accession no. KX377410), *Salmonella enterica serovar* Typhimurium pMCR16_P053 (Genbank accession no. KY352406), and *Citrobacter braakii* pSCC4 (Genbank accession no. CP021078), which are obtained from diverse sources such as fish products, hospital sewage, chickens, and humans. Moreover, it also showed high homology (>99%) to IncP plasmid pHNFP671 (Genbank accession no. KP324830) which carries the *cfr* gene and was isolated from swine feces in Guangdong Province, China. A comparative analysis of the plasmid pHNEP124 and other IncP plasmids indicated that IncP type plasmids have a conserved backbone structure, except plasmid pSCC4, which is missing an ~8.4 kb conjugative region (Figure 3A). In addition, plasmid pHNFP671, with ~35.5-kb insertion, contained a conjugative region and a *cfr*-carrying module (Figure 3A).

**Although the IncP plasmid possessed a conserved backbone region, the variant regions were distinct.** In plasmid pHNEP124, the first mosaic variant regions (~12.5-kb) contained two modules that carried the multi-resistance gene *cfr*, colistin resistance gene *mcr-1*, β-lactams resistance gene *bla*_TEM-1B_, and putative bleomycin resistance gene *ble*. The *cfr*-bearing modules (∆Tn*2*-IS*26*-*ble*-*orf*-∆IS*Enca1*-IS*26*-*cfr*-IS*26*-∆Tn*2*) was inserted in an open reading frame (*orf*) located downstream of *higA* and flanked by 5-bp (TATTT) direct repeats (DRs). In addition, the 3680 bp genetic structure (∆Tn*2*-IS*26*-*ble*-*orf*-∆IS*Enca1*) shared a 99% identity to the corresponding region in the *E. coli* IncHI2 plasmid pLD22-MCR1 (CP047877) and pHNYJC8 (KY019259), which were recovered from deer feces and chicken meat in China, respectively (Figure 4). The *mcr-1*-carrying module (IS*Apl1*-*mcr-1*-∆*pap2*-∆IS*Apl1*) was inserted downstream of the *cfr*-bearing modules and produced 2-bp (GT) DRs, which is in agreement with the *mcr-1*-carrying plasmid pHNGDF36-1 and pMCR_1511. Notably, the *pap2* gene missed the stop codon and has a 7-bp (TAAACTT) homology sequence with delta IS*Apl1* that may be caused by recombination (Figure 4). Another variant region (IS*26*-*hp*-∆IS*6100*) was inserted into an *orf* gene and is flanked by 8-bp (TGAAATTC) DRs, which were also found in the same locus of plasmid pHNGDF36-1 and pSCC4, respectively.

### 2.6. Plasmid Analysis of pHNEP129

The sequence analysis revealed that plasmid pHNEP129 is 35336 bp and belongs to the IncX4 plasmid, which is comprised of a typical IncX4 plasmid backbone including the encoding replication initiation protein gene *pir*, plasmid partitioning protein gene *parA*, encoding DNA relaxase gene *taxC*, a toxin-antitoxin system *hicA-B*, conjugative transfer protein gene *trbM*, and the *virB* gene. BLASTn analysis showed that plasmid pHNEP129 shares a >99% identify with 96% coverage to other *mcr-1*-carrying IncX4 plasmids such as plasmid pMCR1-NY (CP019908, *E. coli*, human) from the United States, plasmid pmcr1_IncX4 (KU761327, *Klebsiella pneumoniae*, human), plasmid pGZ49260 (MG210937, *E. coli*, human), and plasmid pHNSHP10 (MF774182, *E. coli*, swine) isolated from China (Figure 3B). Unlike those *mcr-1*-carrying IncX4 plasmids, plasmid pHNEP129 showed a 97–99% identity with 74% coverage to other *cfr*-bearing IncX4 plasmids such as *E.coli* plasmids pSD11 (KM212169), pGXEC3 (KM580532), pGXEC6 (KM580533), and pEC14cfr (KY865319) recovered from swine farms in different geographic locations of China (Guangdong, Guangxi and Liaoning province; Figure 3B). 

A comparative analysis of the plasmid pHNEP129 and other IncX4 plasmids carried the *mcr-1* or *cfr* genes, the backbone region of plasmid pHNEP129, except for the *cfr*-carrying module, was almost identical to the *mcr-1*-carrying plasmids such as plasmid pGZ49260 and pHNSHP10. However, the backbone region of the plasmid pHNEP129, except for the conjugative transfer region, showed a lower identity to the *cfr*-carrying plasmids such as pSD11, pGXEC3s and pEC14cfr (Figure 3B). In addition, the *cfr*-carrying module contained a *cfr* gene and two copies of IS*26* in the same orientation located downstream of *hns* in plasmid pHNEP129, which was consistent with plasmid pSD11, pGXEC3, and pEC14cfr (Figure 3B). However, the *cfr*-carrying module (*tnpA*-IS*26*-*cfr*-IS*26*) of pGXEC3 and pGXEC6 was inserted in the IncX4 plasmid backbone with 7-bp (TAAAAAC) DRs, but no DRs were found in plasmid pHNEP129. This result indicates that the *cfr*-carrying module (IS*26*-*cfr*-IS*26*) is likely to be directly inserted into the *mcr-1*-positive IncX4 plasmid rather than evolving from *cfr*-positive plasmids such as pSD11 and pGXEC3.

The *cfr*-carrying module (2875-bp) showed a >99% identity to the *cfr*-positive plasmid pGXEC3 and pGXEC6. It is interesting to note that the *cfr*-carrying module has a 43 bp insertion upstream of the *cfr* gene and a 355 bp deletion downstream of the *cfr* gene compared with plasmid pGXEC3 and pSD11 (Figure 4).

## 3. Discussion

Food-producing animals, considered to be a “reservoir” of resistant genes, play a vital role in the dissemination of important resistance genes, such as the *mcr-1* gene. The multiresistant *cfr* gene can confer a broad range of antibiotics resistance (exhibiting PhLOPS_A_ phenotype) in Gram-positive bacteria, but it only mediated resistance to phenicols such as florfenicol in Gram-negative bacteria. In this study, the prevalence (13.7%) of the *cfr* gene in *E. coli* isolates was significantly higher than the previous reported 0.08–1.6% in different provinces of China [8,13,14], which suggested that the *cfr* gene may rapidly disseminate in a specific swine farm at a small scale but not in large areas of China. The *cfr* gene was detected in different sources, which suggests that the *cfr* gene had a high prevalence of circulation in a typical swine farm. To the best of our knowledge, this is the first report on the rapid increase of the *cfr* gene in *E. coli* and the first detection from environmental samples such as soil and sewage in a typical swine farm of Guangdong Province, China.

Compared with the detectable rate of the *cfr* gene in *E. coli* isolates from different growth stages, the higher prevalence of the *cfr* gene in *E. coli* isolates from nursery pigs and fattening pigs may be explained by the long-term use or overuse of antibacterial drugs such as florfenicol, amoxicillin, and gentamycin. Although florfenicol has not been used since 2017 in this swine farm, the detectable rate of the *cfr* gene was obviously increased in *E. coli* isolates, which may because the effects of florfenicol are difficult to eliminate in the short term. A similar result was observed in the spread of the *mcr-1* gene in a swine farm in Shanghai, China [15]. Interestingly, the prevalence of the *mcr-1* gene among *cfr*-positive *E. coli* had decreased sharply from 86.4% before 2017 to 17.1% in 2017, which may be due to the ban of colistin as a feed additive for animals in China in 2016 [16], which is in accordance with the prevalence of the *mcr-1* gene in *E. coli* isolates from a swine farm in Guangdong Province, China [17]. 

The PFGE analysis of all *cfr*-bearing *E. coli* isolates showed that most of these isolates were epidemiologically unrelated. This result suggests that the spread of the *cfr* gene was mainly due to horizontal transfer, which is in agreement with the previous reports [8,12,14]. Previous studies have suggested that the *cfr* gene is mainly located on the plasmid of 23 *E. coli* isolates and on the chromosomal DNA of isolate FSEC-02, and that the IS*26* element plays a vital role in the dissemination of the *cfr* gene [4,18]. The *cfr*-carrying module (IS*26*-*cfr*-IS*26*) was the most prevalent genetic surrounding in *E. coli* isolates except for one *cfr*-carrying module composed of a *cfr* gene and two copies of IS*256*, which was identified in the IncA/C plasmid pSCEC2 from porcine *E. coli* isolates [19]. In addition, two copies of IS*26* in the same orientation have been described to form minicircles comprising the *cfr* gene and one copy of the IS*26* element, which may accelerate the transfer of *cfr* gene by IS*26*-mediated recombination [20], but we failed to detect the minicircles in the current study (data not provided). Furthermore, compared with the previous reports about the genetic surrounding of the *cfr* gene, the *cfr*-carrying module in this study was distinct from the *cfr*-carrying module of plasmid pHNEP28 found in 2013 in this swine farm. Based on this, we assumed that the *cfr*-carrying module has contributed to the dissemination among *E. coli* isolates through the missing delta *rep* gene. Thus, the results provide compelling evidence that plasmids and insertion sequences such as IS*26* are closely related to the spread and diffusion of the *cfr* gene in this swine farm.

The *cfr* gene is mainly found on various Inc type plasmids, including IncX4, IncA/C, IncF43:A-:B-, and IncF14:A-:B-, as well as untyped plasmids in *E. coli* isolates from food-producing animals in China [8]. Plasmids such as IncX4, IncI2, IncP, and IncHI2, which were considered to be the important vectors of the *mcr-1* gene, play an important role in the global spread of the *mcr-1* gene [10]. In the present study, *cfr* and *mcr-1* were first detected in the existence in multiple plasmids such as IncX4, IncI2, and IncP. Notably, the IncP plasmid is a broad-host-range type of plasmid, which had been detected in various sources such as fish products, pig feces, hospital sewage, and humans in China [21,22,23]. It is considered an important carrier of the *mcr-1* and *mcr-3* genes, with the potential to mediate the spread of *mcr-1* or *mcr-3* genes from *Enterobacteriaceae* to other Gram-negative bacteria, such as *Pseudomonas aeruginosa* and *Aeromonas* spp. A previous study reported that *mcr-1*-bearing IncP plasmids have high conjugative frequencies and low fitness costs in hosts, which may facilitate the dissemination of *mcr-1* among various hosts [23]. The IncP plasmid carried multiple resistance genes such as *bla*_TEM-1B_, *ble*, *cfr*, and *mcr-1* genes compared with other IncP plasmids, but this needs to be investigated further in animal husbandry in order to prevent the dissemination of antibiotic-resistant genes. In addition, the *cfr* gene in plasmids pGXEC3 and pSD11 exhibited resistance to florfenicol but most of the plasmids in this study did not. Although the phenomenon of a “silent” *cfr* gene has been described in *Enterococcus faecium* isolates from swine and the patient, this has not been observed in *E. coli* before [24,25], and the mechanism of a “silent” *cfr* in *E. coli* was unknown in this study and needs further research in the future.

## 4. Materials and Methods 

### 4.1. Detection of Multi-Resistant cfr Gene and Other Resistance Genes

A total of 861 samples including 846 anal swabs samples and 14 environmental samples, were collected from the swine farm from May 2014 to February 2017 and the information of the sample collection is listed in Appendix A. All of the samples were cultured using LB broth and were incubated with 200 rpm/min at 37 °C for 14 h. The *E. coli* isolate was screened and purified using MacConkey agar plates without antibiotic selection pressure. Then, a non-duplicate colony was selected and identified using Matrix-Assisted Laser Desorption/Ionization Time of Flight Mass Spectrometry (MALDI-TOF-MS) and 16S rRNA sequencing [26]. In this swine farm, all of the pigs were divided into five growth stages according to the age of the pigs including suckling piglets, nursery pigs, fattening pigs, sows, and boars. Common drugs were investigated and are listed in Table 1. The bacteria DNA was prepared through a boiling method and were used for the PCR amplification and sequencing analysis to detect the *cfr* gene using previously designed primers [27]. Considering that the *cfr* gene is likely to co-transfer with other florfenicol resistance genes and colistin-resistant genes in Gram-negative bacteria, the *cfr*-positive *E. coli* isolates were investigated further using PCR for other florfenicol genes and the *mcr-1* gene, and the primers are listed in Appendix A.

### 4.2. Antibiotic Susceptibility Testing 

Minimum inhibitory concentration (MIC) values of all *cfr*-positive *E. coli* isolates were found using an agar dilution method of 16 different drugs, including ampicillin, cefotaxime, cefquinome, ceftazidime, cefoxitin, imipenem, gentamycin, amikacin, apramycin, neomycin, doxycycline, tetracycline, florfenicol, fosfomycin, ciprofloxacin, and sulfamethoxazole/trimethoprim. The MIC values of colistin were determined via the broth microdilution method. MIC values were interpreted according to the document M100 and VET01-S4 of the Clinical and Laboratory Standards Institute [28,29]. Colistin (>2 mg/L) and florfenicol (>16 mg/L) were interpreted according to the clinical breakpoints or epidemiological cut-off values of the European Committee on Antimicrobial Susceptibility Testing (http://mic.eucast.org/Eucast2/). ATCC 25922 served as the control.

### 4.3. Pulsed-Field Gel Electrophoresis (PFGE) and Flanking Regions of the cfr Gene

The PFGE analysis of all *cfr*-positive strains with XbaI-digested genomic DNA was performed using the CHEF-MAPPER System (Bio-Rad Laboratories, Hercules, CA, USA) as described previously [30]. The PFGE patterns were analyzed using BioNumerics software (Applied Maths, Sint-Martens-Latem, Belgium) with a cut-off value at 80% of the similarity values in order to indicate different PFGE patterns according to a previous report [14]. The flanking regions of the *cfr* gene in the *E. coli* isolates were determined by PCR mapping and the sequences of the primers are listed in Appendix A.

### 4.4. S1-PFGE and Southern Blotting Hybridization

S1 nuclease pulsed-field gel electrophoresis combined with Southern blotting hybridization with the *cfr* probe was conducted to determine the location of the *cfr* gene in *E. coli*, as well as the size of the *cfr*-carrying plasmid according to previous protocols [12,15]. 

### 4.5. Conjugation Experiment and Transformation Assay

The conjugation experiment was conducted with the broth mating method using streptomycin-resistant *E. coli* C600 as the recipient strain. The 45 *E. coli* isolates carrying the *mcr-1* and *cfr* genes were used as the donor strains. After 4 h of the culture of the donor strains and *E. coli*, C600 were mixed (ratio of 1:4) in Luria-Bertani (LB) broth, and then put into incubation for 4 h at 37 °C. The transconjugants were selected on LB agar plates supplemented with streptomycin (3000 mg/L) and florfenicol (10 mg/L) or colistin (2 mg/L). To obtain the transferable plasmids co-harboring the *mcr-1* and *cfr* genes, the *cfr/mcr-1*-positive plasmids of the transconjugants were transformed into the *E. coli* DH5α (Takara) through electroporation. Transformants were selected on LB agar plates containing 10 mg/L florfenicol or 2 mg/L colistin. The antimicrobial susceptibility of the transconjugants or transformants was determined by either the agar dilution method or the broth microdilution method, and the presence of *cfr*, *mcr-1*, and *floR* genes in the transconjugants or transformants were identified by PCR. 

### 4.6. PBRT and Plasmid Analysis

PCR-based replicon typing (PBRT) was performed on all *cfr/mcr-1*-positive transconjugants/transformants using the primers as described previously [31,32]. Plasmids pHNEP124 and pHNEP129 bearing the *cfr* and *mcr-1* genes from transformants GDE6P124T and GDE6P129T were extracted and purified using a Qiagen plasmid midi kit (Qiagen, Hilden, Germany), and were subjected to sequencing by Illumina Miseq 2500 (Illumina, San Diego, CA, USA). The sequence reads were assembled into contigs using SOAP denovo version 2.04, and the gaps between the contigs were linked by PCR and sequencing. The complete sequences of plasmids pHNEP124 and pHNEP129 were analyzed and annotated by IS finder (https://www-is.biotoul.fr/), BLASTn (https://blast.ncbi.nlm.nih.gov/Blast.cgi), ResFinder (https://cge.cbs.dtu.dk//services/ResFinder/), RAST [33], and Vector NTI program (Invitrogen, San Diego, CA, USA).

### 4.7. Nucleotide Sequence Accession Number

The nucleotide sequences of plasmids pHNEP124 and pHNEP129 were deposited in the GenBank database under the accession numbers MT667260 and MT667261, respectively.

## 5. Conclusions

This is the first study to report the high prevalence of the *cfr* gene among *E. coli* isolates in a typical swine farm from 2014–2017 and the first identification of the complete genome sequence of IncP and IncX4 plasmids co-harboring the *mcr-1* and *cfr* genes. Florfenicol was extensively used for preventing and treating an infectious disease, which might have facilitated the spread of the *cfr* gene in this swine farm. Plasmids such as IncX4 and insertion sequence IS*26* were responsible for the dissemination of the *cfr* gene in this study. The occurrence and dissemination of plasmids carrying the *cfr* and *mcr-1* genes in *E. coli* isolates need further surveillance.

## Figures and Tables

**Figure 1 pathogens-10-00033-f001:**
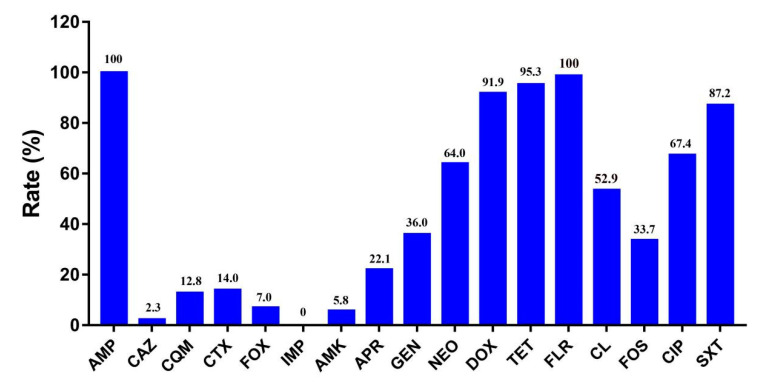
Resistance rate of 85 *cfr*-positive *E. coli* isolates. AMP—ampicillin; CAZ—ceftazidime; CQM—cefquinome; CTX—cefotaxime; FOX—cefoxitin, IMP—imipenem; AMK—amikacin; APR— apramycin; GEN—gentamycin; NEO—neomycin; DOX—doxycycline; TET—tetracycline; FLR—florfenicol; CL—colistin; FOS—fosfomycin; CIP—ciprofloxacin; SXT—sulfamethoxazole/trimethoprim.

**Figure 2 pathogens-10-00033-f002:**
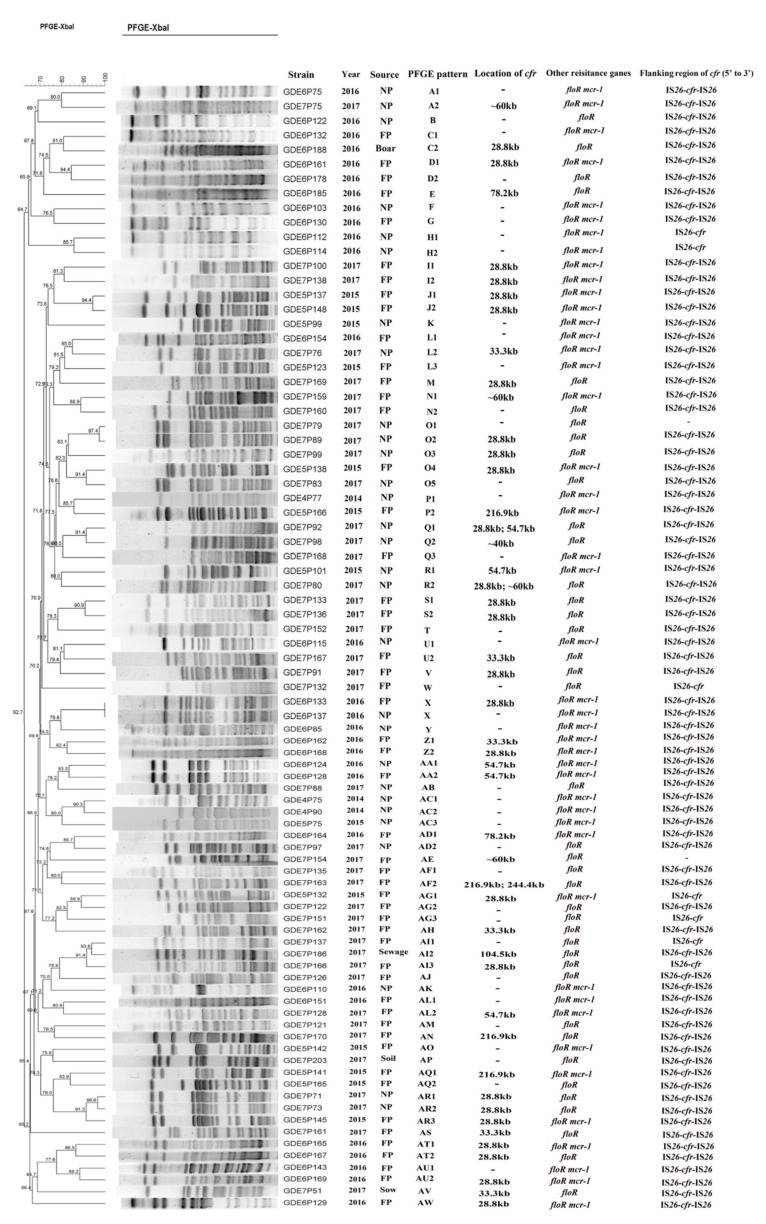
Characterization of 85 *cfr*-positive *E. coli* isolates in this study. NP—nursery pig; FP—fattening pig. “–” indicates none detected.

**Figure 3 pathogens-10-00033-f003:**
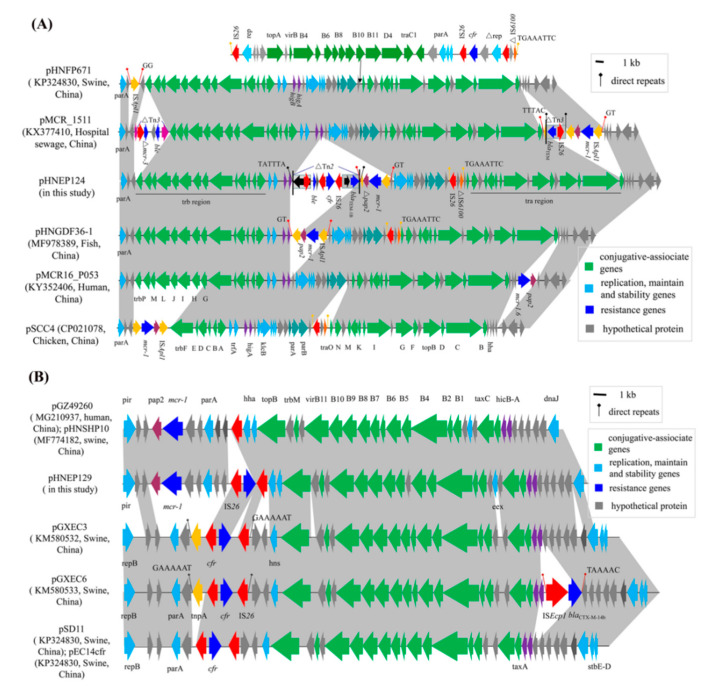
Liner comparison of the *cfr*-carrying plasmids. Arrows indicate the positions of the genes and the direction. Regions with >99% homology are shaded in gray. ∆ indicates a truncated gene or mobile element. (**A**) Comparative analysis of plasmid pHNEP124 and other IncP-type plasmids. (**B**) Comparative analysis of plasmid pHNEP129 and other IncX4-type plasmids.

**Figure 4 pathogens-10-00033-f004:**
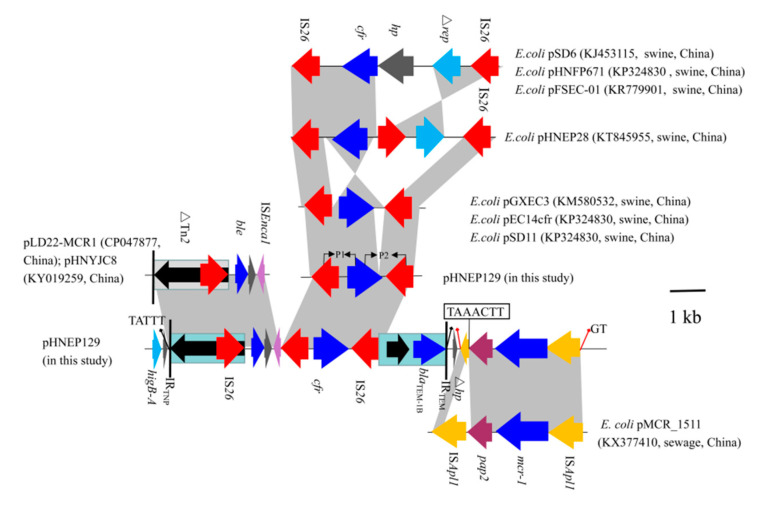
Comparison of the genetic context of *cfr* gene among *E. coli* strains. Arrows indicate the positions of the genes and the direction. Regions with >99% homology are shaded in gray. ∆ indicates a truncated gene or mobile element.

**Table 1 pathogens-10-00033-t001:** Prevalence of the *cfr* gene in *E. coli* isolates from various sources, as well as information on drug use in the swine farm during the period of 2014–2017.

Collected Date	Suckling Piglet (N)	Nursery Pig (N)	Fattening Pig (N)	Sow (N)	Boar (N)	Environment (N)	Total	History of Drug Use
Jun. 2014	33	35 (3)	64	52	0	0	184 (3, 1.6%)	gentamycin, amoxicillin
Jul. 2015	28	39 (3)	46 (10)	38	0	0	151 (13, 8.6%)	florfenicol
Jul. 2016	8	32 (9)	45 (18)	51	4 (1)	1	141 (28, 19.9%)	florfenicol, enrofloxacin, mequindox, kanamycin
Apr. 2017	14	23 (14)	51 (24)	36 (1)	9	8 (2)	141 (41, 29.1%)	gentamycin, amoxicillin, kanamycin
Total	83 (0)	129 (29, 22.5%)	206 (52, 25.2%)	177 (1, 0.6%)	13 (1, 7.7%)	9 (2, 22.2%)	617 (85, 13.7%)	-

Note: the number and detection rate of the *cfr*-positive *E. coli* isolates are listed in brackets.

**Table 2 pathogens-10-00033-t002:** Characterization of 30 transconjugants carrying the *mcr-1* and *cfr* genes.

Strains	MIC Values		Replicon Type	Resistance Phenotype ^a^	Resistance Genes
	Florfenicol	Colistin			
GDE5P101J	2	4	IncI2	CL	*cfr mcr-1*
GDE5P123J	4	4	IncI2	AMP CL	*cfr mcr-1*
GDE5P132J	2	4	IncX4	CL	*cfr mcr-1*
GDE5P137J	4	8	IncX4	AMP APR CL	*cfr mcr-1*
GDE5P138J	4	4	IncX4	CL	*cfr mcr-1*
GDE5P141J	4	8	IncHI2	CL	*cfr mcr-1*
GDE5P142J	8	4	IncX4	CL	*cfr mcr-1*
GDE5P145J	2	8	IncX4	CL	*cfr mcr-1*
GDE5P148J	8	4	-	AMP CL	*cfr mcr-1*
GDE5P165J	8	4	IncX4	CL	*cfr mcr-1*
GDE6P75J	8	4	IncHI2	AMP CTX CL FOS	*cfr mcr-1*
GDE6P85J	4	4	IncX4	CL	*cfr mcr-1*
GDE6P103J	4	4	-	AMP CL	*cfr mcr-1*
GDE6P110J	8	2	IncX4	CL	*cfr mcr-1*
GDE6P115J	2	4	-	AMP CL	*cfr mcr-1*
GDE6P124J	8	4	IncP ^b^	AMP CL	*cfr mcr-1*
GDE6P128J	8	4	IncP	AMP CL	*cfr mcr-1*
GDE6P129J	4	4	IncX4	CL	*cfr mcr-1*
GDE6P130J	4	4	IncX4	AMP CL	*cfr mcr-1*
GDE6P133J	>128	4	IncX4	AMP FLR CL	*cfr mcr-1 floR*
GDE6P143J	4	4	IncX4	CL	*cfr mcr-1*
GDE6P151J	>128	4	IncX4	AMP GEN FLR CL	*cfr mcr-1 floR*
GDE6P164J	4	4	IncX4	CL	*cfr mcr-1*
GDE6P165J	4	4	IncX4	CL	*cfr mcr-1*
GDE6P168J	4	4	IncX4	AMP CL	*cfr mcr-1*
GDE6P169J	4	4	IncX4	AMP CL	*cfr mcr-1*
GDE7P75J	16	4	IncX4	CL SXT	*cfr mcr-1*
GDE7P100J	8	4	IncHI2 IncX4	AMP APR DOX CL SXT	*cfr mcr-1*
GDE7P128J	8	8	IncI2	AMP CL SXT	*cfr mcr-1*
GDE7P166J	16	2	IncX4	AMP DOX CL	*cfr mcr-1*
*coli* C600	4	0.125	- ^c^	-	-
ATCC 25922	2	0.5	-	-	-

Note: (^a^) AMP—ampicillin; CTX—cefotaxime; APR—apramycin; GEN—gentamycin; FLR—florfenicol; DOX—doxycycline; CL—colistin; FOS—fosfomycin; SXT—sulfamethoxazole/trimethoprim. (^b^) The replicon type of IncP was detected using the primer list in Appendix A. (^c^) “–” indicates none detected.

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
