# Peer review of "Rapid Increase in the IS26-Mediated cfr Gene in E. coli Isolates with IncP and IncX4 Plasmids and Co-Existing cfr and mcr-1 Genes in a Swine Farm"

_pathogens, 2021, doi:10.3390/pathogens10010033_

Round 1
Reviewer 1 Report
This manuscript presents the analysis of antimicrobial resistance found in isolates of E. coli found on a single swine farm. Select isolates were further analyzed for resistance phenotypes associated with the cfr gene and the mcr-1 gene. DNA sequencing was preformed to identify the genetics of these resistance genes. Several were found to be located on plasmids and the cfr-1 gene was linked to is26. I would suggest that the authors consider adding the figure in the Supplemental data (Figure S1) to the manuscript with a brief description of the assumed evolution of the mobile elements. While these are interesting findings, they are of a study of a single swine farm and may be difficult to interpret on a larger scale of the swine industry. The rapid increase in the prevalence of the cfr gene on this farm is interesting it would benefit from comparison to other similar farms. Also, the paper needs to be proof read by a native English speaker as their are numerous errors for example:
line 54, "no" versus "know"
line 58 "had" rather than "with"
Author Response
Response 1: Thanks for your helpful suggestions. Based on your comment, the Figure S1 converting into Figure 4 has been inserted into the manuscript, and the evolution of cfr gene was described in Line 246-250. A revised manuscript with the correction sections was marked with blue font. In addition, English language and grammar of the manuscript has been revised by the MDPI author services. We hope the revised manuscript can meet your requirement.
Reviewer 2 Report
In this manuscript, the authors describe a number of E. coli isolates which carry the resistance gene cfr. Isolates were obtained from a single pig farm over a 3 year period. A total of 617 isolates were obtained from six sample types (5 of which represented different pig growth/production stages).
PCR screen was carried out for selected antibiotic resistances, with a focus towards cfr, floR and mcr resistance genes. Investigative priority was given to 85 isolates positive for cfr, among which a subset carried floR and/or mcr (specifically identified as mcr-1).
Diversity of selected isolates was assessed by PFGE.
Conjugative transfer was achieved for 30 of 45 isolates positive for both cfr and mcr. Resistance phenotype was also characterised. Two plasmids carried cfr, mcr and floR genes
Two plasmids were subjected to sequencing and subsequent comparative analysis with similar plasmids. cfr was associated with IS26 element and this was adjacent to mcr – potential for co-selection was noted.
The emergence and prevalence of cfr and its co-association with mcr is of interest and has implications for further spread.
COMMENTS
- Some further editing of the manuscript is required.
- Further editing is required to remove guidance notes (lines 71-73)
- Analyses and presentation of results seems appropriate for the purpose.
- Overall, methodologies seem to be appropriate and described adequately. However, the method for isolation of E. coli from samples (e.g. selective/indicator media) is not stated.
- From all of the sampling it appears that only 617 E. coli isolates were obtained. Was selection based on one colony per sample? Whether this or another selection method was used should be stated in methodology.
- Identifying from which isolate and/or conjugant each of the 2 plasmids originated should be stated.
Author Response
Point 1: Some further editing of the manuscript is required.
Response 1: Thanks for your suggestion. A revised manuscript with the correction sections was marked with blue font. Based on your comment, we has delivered the manuscript to MDPI author services for English editing. Thus, we hope the revised manuscript can meet your requirement.
Point 2: Further editing is required to remove guidance notes (lines 71-73)
Response 2: Thanks for your suggestion.We has remove the guidance notes in Line 71-73.
Point 3: Analyses and presentation of results seems appropriate for the purpose.
Response 3: Thanks for your kindness recommend.
Point 4: Overall, methodologies seem to be appropriate and described adequately. However, the method for isolation of E. coli from samples (e.g. selective/indicator media) is not stated.
Response 4: Thanks for your helpful suggestion. Based on your comment, we have supplied the methodology of E. coli isolation in Line 258-264 and the information of sample collection in Table S1.
Point 5: From all of the sampling it appears that only 617 E. coli isolates were obtained. Was selection based on one colony per sample? Whether this or another selection method was used should be stated in methodology.
Response 5: Thanks for your helpful suggestion. A total of 861 samples including 846 anal swabs samples and 14 environmental samples, were collected from the swine farm from May-2014 to Feb-2017. E. coli isolate was screened and purified using MacConkey agar plates without antibiotic selection pressure. And then a non-duplicate colony was selected and identified using MALDI-TOF-MS and 16S rRNA sequencing. we have supplied the methodology of E. coli isolation in Line 258-264.
Reviewer 3 Report
Revision of manuscript pathogens-1048420
Dear Authors,
Your manuscript entitled “Rapidly Increase of IS26-mediated cfr Gene Among E. coli isolates with IncP and IncX4 Plasmids Co-existing cfr and mcr-1 Gene in a Swine Farm” is a very interesting work on cfr and mcr-1 genes distribution, co-occurrence and correlation. Analysis of the same farm for a long period and considering all swine categories help to have information about in-farm epidemiology of these genes. The work is well planned and conducted. Results were clearly presented and discussed. Overall, it is a very good work. However, I have 2 major and some minor suggestions for Authors:
- It seems (I don’t know if it is true, but this is my impression) that the manuscript come form a deep revision. Please, check carefully all the manuscript because some parts are not very clear.
- Check all bacteria and gene names for italic.
Below some minor corrections:
- Introduction
- Line 39: cfr in italic.
- Line 55: E. coli in italic.
- Line 58: cfr in italic.
- Matherial and Methods:
- Lines 266-267: bacteria and genes in italic; “Gram-negative”.
- Lines 259-267: a more detailed explanation on sampling (time, rhythm, tipe of samples, animals sampled and methods) and on isolate detection and selection is necessary.
- Line 269-273: references for methods should be added.
- Lines 286-288: add a reference or better describe this method.
- Line 293: delete last “incubator”.
- Results
- Line 64: E. coli in italic.
- Line 69: bacteria and genes in italic.
- Line 68-71: this is a discussion on obtained results, not a result.
- Line 71-73: Some error occurred.
- Line 92: delete “in”.
- Line 100, 104 and 115: E. coli in italic.
- Line 127: mcr-1 and cfr in italic.
- Line 146-147: Klebsiella pneumoniae, Salmonella enterica and Citrobacter braakii in italic
- Discussion:
- Line 209: “can confers a broad range of antibiotics” à “can confers a broad range of antibiotics resistance”.
- Line 241: mcr-1 in italic.
- Line 249: delete one “may”.
- Line 255: “swine and the patient” this is not clear.
I sincerely hope that these suggestions will enhance this manuscript. However, if I have made any errors or misinterpretations, I apologize in advance.
Sincerely
The Reviewer
Author Response
Point 1: It seems (I don’t know if it is true, but this is my impression) that the manuscript come form a deep revision. Please, check carefully all the manuscript because some parts are not very clear.
Response 1: Thanks for your kindness comment on our manuscript. In fact, we had carefully checked the manuscript before we submitted it. Now, English language and grammar of the manuscript was revised and polished by the MDPI author services. A revised manuscript with the correction sections was marked with blue font. We hope the revised manuscript can meet your requirement.
Point 2: Check all bacteria and gene names for italic.
Below some minor corrections:
- Introduction
- Line 39: cfr in italic.
- Line 55: E. coli in italic.
- Line 58: cfr in italic.
- Matherial and Methods:
- Lines 266-267: bacteria and genes in italic; “Gram-negative”.
- Lines 259-267: a more detailed explanation on sampling (time, rhythm, tipe of samples, animals sampled and methods) and on isolate detection and selection is necessary.
- Line 269-273: references for methods should be added.
- Lines 286-288: add a reference or better describe this method.
- Line 293: delete last “incubator”.
- Results
- Line 64: E. coli in italic.
- Line 69: bacteria and genes in italic.
- Line 68-71: this is a discussion on obtained results, not a result.
- Line 71-73: Some error occurred.
- Line 92: delete “in”.
- Line 100, 104 and 115: E. coli in italic.
- Line 127: mcr-1 and cfr in italic.
- Line 146-147: Klebsiella pneumoniae, Salmonella enterica and Citrobacter braakii in italic
- Discussion:
- Line 209: “can confers a broad range of antibiotics” à “can confers a broad range of antibiotics resistance”.
- Line 241: mcr-1 in italic.
- Line 249: delete one “may”.
- Line 255: “swine and the patient” this is not clear.
Response 2: Thanks for your suggestions. All bacteria and gene names were carefully rechecked in this manuscript. Based on your comment, we have supplied the methodology of E. coli isolation in Line 275-282 and the information of sample collection in Table S1. References for methods had been added in Line 306 and Line 312. In addition, we remove Line 68-71 to Line 212-216 and delete “swine and the patient” . We hope the response can meet your requirement.
This manuscript is a resubmission of an earlier submission. The following is a list of the peer review reports and author responses from that submission.